# Say One Thing, Do Another? Diagnosing Reasoning-Execution Gaps in VLM-Powered Mobile-Use Agents

## Abstract

Mobile-use agents powered by vision-language models (VLMs) have shown great potential in interpreting natural language instructions and generating corresponding actions based on mobile graphical user interface. Recent studies suggest that incorporating chain-of-thought (CoT) reasoning tends to improve the execution accuracy. However, existing evaluations emphasize execution accuracy while neglecting whether CoT reasoning aligns with ground-truth actions. This oversight fails to assess potential reasoning-execution gaps, which in turn foster over-trust: users relying on seemingly plausible CoTs may unknowingly authorize harmful actions, potentially resulting in financial loss or trust crisis. In this work, we introduce a new evaluation framework to diagnose reasoning-execution gaps. At its core lies Ground-Truth Alignment (GTA), which measures whether the action implied by a CoT matches the ground-truth action. By combining GTA with the standard Exact Match (EM) metric, we jointly assess both the reasoning accuracy and execution accuracy. This joint perspective reveals two types of reasoning-execution gaps: (i) Execution Gap (EG), where the reasoning correctly identifies the correct action but execution fails, and (ii) Reasoning Gap (RG), where execution succeeds but reasoning process conflicts with the actual execution. Experimental results across a wide range of mobile interaction tasks reveal that reasoning-execution gaps are prevalent, with execution gaps occurring more frequently than reasoning gaps. Moreover, while scaling up model size reduces the overall gap, sizable execution gaps persist even in the largest models. Further analysis shows that our framework reliably reflects systematic EG/RG patterns in state-of-the-art models. These findings offer concrete diagnostics and support the development of more trustworthy mobile-use agents. Our data and code are publicly available at *anonymity*.

## 1 Introduction

Mobile-use agents powered by vision-language models (VLMs) are increasingly capable of following natural language instructions and operating graphical user interfaces (GUIs) on mobile devices. By interpreting screenshots and mapping them to executable actions, these agents hold promise for tasks such as app navigation, automation, and accessibility support (Wang et al., 2024c; Nguyen et al., 2024; Zhang et al., 2024a; Liu et al., 2025a; Hu et al., 2025). Recent studies further show that incorporating chain-of-thought (CoT) reasoning can enhance execution accuracy, as intermediate reasoning steps help models decompose complex instructions and align with user intent (Zhang et al., 2025f; Qin et al., 2025; Ye et al., 2025; Zhang et al., 2025b).

Despite these advances, current evaluation practices remain limited. The dominant metric, Exact Match (EM), checks whether the predicted action exactly matches the ground truth (Zhang et al., 2025f; Wang et al., 2025; Shi et al., 2025). However, EM alone overlooks whether the CoT is consistent with the target action, obscuring cases where the agent acts correctly with the wrong reason or fails despite plausible reasoning. Such reasoning-execution gaps pose risks for over-trust, complicate debugging, and undermine the reliability of these systems (Barez et al., 2025; Zhao et al., 2025; Chen et al., 2025a; Matton et al., 2025).

This gap highlights the need for a principled evaluation framework that explicitly accounts for reasoning-execution gaps. A key challenge is to determine whether a CoT faithfully implies the ground-truth action and to disentangle whether errors arise from flawed reasoning or faulty execution. Without such diagnostics, existing metrics mix distinct error sources, offering limited insight into model behavior (Shi et al., 2025; Tang et al., 2025).

To address this challenge, we introduce Ground-Truth Alignment (GTA), a new metric that evaluates whether the action implied by the CoT matches the ground-truth action. By combining GTA with the conventional EM metric, we derive a four-quadrant diagnostic framework that categorizes model outputs into: (i) Ideal, where both reasoning and action correct, (ii) Execution Gap (EG), where reasoning is correct but execution fails, (iii) Both Wrong, where both reasoning and action are incorrect, and (iv) Reasoning Gap (RG), where the action is correct but reasoning is inconsistent. This framework provides fine-grained insights into where and why models fail.

Our contributions can be summarized as follows:

(i) We introduce GTA, a principled metric that measures whether the action implied by an agents CoT aligns with the ground-truth action. By combining GTA with the standard EM metric, we derive a four-quadrant diagnostic space that disentangles reasoning accuracy from execution accuracy and reveals distinct error modes.

(ii) We develop an automatic GTA Evaluator that translates free-form CoT reasoning into a GTA score. Using a stratified sample of human annotations, we confirm that the evaluator offers reliable and consistent assessments, enabling large-scale reasoning diagnostics without expensive manual labeling.

(iii) We conduct extensive experiments on diverse mobile-interaction benchmarks, including AITZ (Zhang et al., 2024b), CAGUI (Zhang et al., 2025f), and AndroidControl (Li et al., 2024), systematically quantify and characterize two key failure modes: EG and RG. Our results show that reasoning-execution gaps are widespread and that EG occurs more frequently, even in the largest state-of-the-art VLM agents.

## 2 RELATED WORK

In this section, we first review the development of mobile-use agents and discuss the recent progress in leveraging CoT reasoning to improve task performance. Then, we dive into the studies of the faithfulness of mobile-use agents that forms the basis of this work.

### 2.1 MOBILE-USE AGENTS

Mobile-use agents aim to operate smartphone applications autonomously, perceiving dynamic GUI states and performing fine-grained actions such as taps, swipes, and text input. Recent advances in VLMs (Wang et al., 2024b; Bai et al., 2025; Yao et al., 2024; OpenAI, 2024; 2025; Comanici et al., 2025; Team et al., 2024) have enabled agents to process screenshots and natural language instructions directly. These developments have sparked intensive research on how to improve reasoning (Zhang et al., 2025f; Ye et al., 2025; Zhang et al., 2025c), grounding (Wu et al., 2024; Cheng et al., 2024; Zhou et al., 2025; Gou et al., 2024), and reliability (Cheng et al., 2025a) in mobile-use agent. According to Mobile-Agent-v3 (Ye et al., 2025), existing mobile-use agents can be divided into two categories: single-agent and multi-agent system.

Single-agent methods usually follow an agent-as-a-model paradigm (Chen et al., 2024), where a single VLM is trained through continue pretraining (CPT), supervised fine-tuning (SFT), and reinforcement learning (RL) to jointly handle perception, reasoning, and action prediction. Early works (e.g., UGround (Gou et al., 2024), OS-Atlas (Wu et al., 2024), CogAgent (Hong et al., 2024), UI-TARS (Qin et al., 2025)) demonstrate that combining GUI-specific pretraining with SFT on mobile interaction data yields competitive results. More recent research focuses on reinforcement fine-tuning (RFT) strategies, such as GRPO, to enhance reasoning ability. Representative RFT-based systems include UI-R1 (Lu et al., 2025), GUI-R1 (Luo et al., 2025), InfiGUI-R1 (Liu et al., 2025b), AgentCPM-GUI (Zhang et al., 2025f), GUI-OWL (Ye et al., 2025), MagicGUI (Tang et al., 2025), UI-Venus (Gu et al., 2025b), Mobile-R1 (Gu et al., 2025a), and BTL-UI (Zhang et al., 2025c).

Multi-agent system decomposes mobile-use agents into specialized roles. For example, planner-executor frameworks (Wang et al., 2024a; Xu et al., 2025b; Guo et al., 2025),separate high-level task decomposition from low-level interaction; memory-augmented systems (Zhang et al., 2025a; Wang et al., 2024d; Agashe et al., 2025) enhance long-horizon consistency; and reflection-based designs (Li et al., 2025; Zheng et al., 2024) iteratively refine decisions before execution.

Unlike prior work focused on execution accuracy, our study investigates whether an agents internal reasoning process faithfully supports its predicted actions. We introduce an evaluation framework to diagnose reasoning-execution gaps. This perspective provides a new lens to analyze mobile-use agents, complementing existing benchmarks and highlighting overlooked reliability issues.

## 2.2 MOBILE-USE AGENTS WITH CoT

Given the proven success of Chain-of-Thought reasoning in large language models (LLMs) (Zhang et al., 2025e; Xu et al., 2025a; Cheng et al., 2025b;c), researchers have recently extended this paradigm to VLM-powered agents. By explicitly modeling intermediate reasoning steps, CoT enhances execution accuracy in complex tasks. In the domain of mobile-use agents, CoT has shown notable benefits: it boosts execution accuracy, strengthens human-agent interaction, and organizes historical context, thereby becoming central to modern GUI automation systems (Zhang et al., 2025b; Qin et al., 2025; Ye et al., 2025; Liu et al., 2025b; Luo et al., 2025).

The CoT training paradigm evolved from SFT to RFT. Early systems integrated CoT into mobile-use agents through SFT (Zhang & Zhang, 2023; Zhang et al., 2024b; Hong et al., 2024). However, SFT depends on large amounts of expert-annotated data and shows weak robustness when applied to out-of-distribution tasks. To address these issues, recent work adopts RFT (Zhang et al., 2025f; Liu et al., 2025b; Tang et al., 2025; Luo et al., 2025; Lu et al., 2025; Zhang et al., 2025c; Gu et al., 2025a), to better elicit reasoning ability of mobile-use agents.

Despite these advances, evidence shows that the current mobile-use agents struggles with reasoning-execution gaps (Gu et al., 2025b). However, there is still a lack of studies on analyzing and understanding the challenge. This paper fills the gap by proposing a novel metric that disentangles reasoning-execution gaps into two distinct components: the execution gap and the reasoning gap. We conduct a systematic analysis of state-of-the-art models using this metric, offering new insights into their performance and failure modes.

## 2.3 FAITHFULNESS OF MOBILE-USE AGENTS

With the increasing deployment of mobile-use agents across a wide range of real-world applications, a fundamental open question remains: can these agents remain faithful in complex, dynamic environments (Zhang et al., 2024c; Ma et al., 2025; Shi et al., 2025)? The challenge of faithfulness in such agents stems from two key dimensions: faithfulness to user intentions and faithfulness to the agents own decision-making process.

Regarding the first dimension, recent studies have shown that mobile agents are highly susceptible to environmental distractions, such as pop-up boxes (Ma et al., 2025; Zhang et al., 2025d; Chen et al., 2025b). These distractions can derail agents from pursuing user-specified goals and may lead them into unintended or even unsafe environmental states. Interestingly, applying CoT (e.g., instructing agents to pay attention to potential distractions) does not alleviate the issue, but even increases the likelihood of distraction. This suggests that naive application of CoT may not inherently improve agent faithfulness.

For the second dimension, prior work has explored methods for measuring the confidence of decision making process (Cheng et al., 2025a; Hao et al., 2025; Tao et al., 2025), thus allowing agents to proactively query users in uncertain or potentially harmful situations (Wu et al., 2025; Ai et al., 2025; Cheng et al., 2025d). For example, VeriOS-Agent (Wu et al., 2025) is capable of identifying untrustworthy scenarios, asking appropriate clarification queries, and following clarified instructions in such settings without degrading performance in routine tasks.

Despite these advances, there remains a gap in foundational understanding of reasoning-execution gaps, which underpins both aspects of agent faithfulness. This paper addresses this gap by introducing a new diagnostic framework with fine-grained metrics designed to systematically quantify

the reasoning-execution gaps. Through this, we aim to deepen understanding of mobile-use agent behavior and support the development of more reliable agents suitable for real-world deployment.

# 3 METHOD

In this section, we first introduce the VLM-based mobile agent and formalize the task. We then present our evaluation metrics, Exact Match (EM) for execution accuracy and Ground-Truth Alignment (GTA) for reasoning accuracy, along with a diagnostic framework. Finally, we describe the design of the GTA Evaluator, a key component for assessing reasoning quality.

## 3.1 VLM-BASED MOBILE AGENT

This paper investigates VLM-based mobile agents, which generate CoT reasoning and executable actions directly from natural language instructions and GUI screenshots. Formally, the end-to-end task is defined as modeling the conditional probability:

$$P(c_n, a_n \mid I, H_n, o_n), \tag{1}$$

where $I$ denotes the task instruction, $o_n$ represents the current screenshot, and $H_n$ denotes optional history context. The representation of $H_n$ varies across agents, depending on their design and training paradigms. For example, AgentCPM-GUI (Zhang et al., 2025f) does not incorporate history, thus treating $H_n$ as empty. In contrast, UI-TARS (Qin et al., 2025) encodes $H_n$ with dialogue-style history format, while GUI-Owl (Ye et al., 2025) compresses the historical context for efficient integration. Detailed descriptions are provided in the Appendix C. Importantly, we focus on models that explicitly generate $(c_n, a_n)$, which enables us to analyze the reasoning-execution gaps.

## 3.2 EVALUATION METRICS

Following standard GUI benchmarks (Zhang et al., 2025f; Wang et al., 2025; Shi et al., 2025), we adopt **Exact Match (EM)** for execution accuracy, and further introduce **Ground-Truth Alignment (GTA)** for reasoning accuracy.

**Exact Match (EM).** Let $a_n^*$ be the ground-truth action at step $n$ and $a_n$ the models executed action. Formally, let $\mathbf{1}_{\{\cdot\}}$ denote the indicator function:

$$\text{EM}_n = \mathbf{1}_{\{a_n=a_n^*\}}, \tag{2}$$

where equality requires both the action type and parameters to match. The overall EM score is defined as the step-level average:

$$\text{EM} = \frac{1}{N} \sum_{n=1}^{N} \text{EM}_n. \tag{3}$$

**Ground-Truth Alignment (GTA).** Let $c_n$ denote the CoT generated at step $n$, and $f(c_n)$ the action implied by the CoT:

$$\text{GTA}_n = \mathbf{1}_{\{f(c_n)=a_n^*\}}, \tag{4}$$

measuring whether the CoT correctly leads to the ground-truth action. The overall GTA score is

$$\text{GTA} = \frac{1}{N} \sum_{n=1}^{N} \text{GTA}_n. \tag{5}$$

**Quadrant Analysis.** Combining EM and GTA yields a four-quadrant categorization (Figure 1):

- Q1: $\text{EM}_n = 1, \text{GTA}_n = 1$ (Ideal, where both reasoning and action correct)
- Q2: $\text{EM}_n = 0, \text{GTA}_n = 1$ (Execution Gap, where reasoning is correct but execution fails)
- Q3: $\text{EM}_n = 0, \text{GTA}_n = 0$ (Both Wrong, where both reasoning and action are incorrect)
- Q4: $\text{EM}_n = 1, \text{GTA}_n = 0$ (Reasoning Gap, where the action is correct but reasoning fails)

**Diagnostic Rates.** Beyond individual metrics, we further define two diagnostic rates to capture mismatch patterns between reasoning and action:

$$\text{EG} = \frac{1}{N} \sum_{n=1}^{N} \mathbf{1}_{\{\text{GTA}_n=1 \wedge \text{EM}_n=0\}}, \quad (6)$$

$$\text{RG} = \frac{1}{N} \sum_{n=1}^{N} \mathbf{1}_{\{\text{GTA}_n=0 \wedge \text{EM}_n=1\}}. \quad (7)$$

Here, **Execution Gap (EG)** quantifies the fraction of steps where the CoT implies the correct action but the executed action is wrong (*thought correct, action wrong*), while **Reasoning Gap (RG)** quantifies the fraction of steps where the executed action is correct but the CoT is wrong (*thought wrong, action correct*).

### 3.3 GTA EVALUATOR

A key challenge in computing GTA lies in mapping the free-form CoT $c_n$ into its implied action $f(c_n)$. Since CoT is often expressed in natural language, we design an automatic evaluator based on an instruction-following VLM. Specifically, we formulate the mapping as

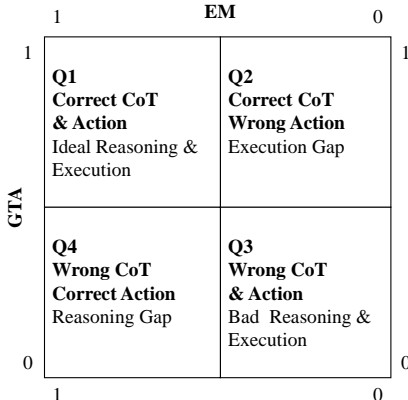

Figure 1: Four-quadrant diagnostic framework of reasoningexecution gaps. The axes represent reasoning accuracy (GTA) and action accuracy (EM). Q1: Ideal, where both reasoning and action correct; Q2: Execution Gap (EG), where reasoning is correct but execution fails; Q3 Both Wrong, where both reasoning and action are incorrect; Q4: Reasoning Gap (RG), where the action is correct but reasoning fails.

$$f(c_n) = \arg\max_a P(a \mid c_n, H_n, o_n), \quad (8)$$

where the evaluator predicts the most likely executable action $a$ given the CoT reasoning $c_n$, the local history $H_n$, and the current observation $o_n$. We enforce deterministic decoding (greedy decoding) to guarantee reproducibility of $f(c_n)$.

We retain $(H_n, o_n)$ in the evaluators input because CoT $c_n$ often contains underspecified or context-dependent references (e.g., "click the confirm button below). Without grounding in the current screenshot $o_n$ and dialogue history $H_n$, the implied action could be ambiguous or even infeasible. Conditioning on $(c_n, H_n, o_n)$ therefore ensures that $f(c_n)$ consistently maps reasoning traces to executable actions in the same action space as the ground-truth annotation.

Finally, the evaluation of GTA follows the same matching rule as EM: both the action type and all parameters must match the ground-truth annotation. This consistent criterion ensures that reasoning accuracy (GTA) and execution accuracy (EM) are directly comparable.

## 4 EXPERIMENT

In this section, we conduct comprehensive experiments to address the following research questions:

- **RQ1:** To what extent does the proposed GTA Evaluator provide reliable and reproducible measurements of reasoning accuracy?
- **RQ2:** How do state-of-the-art VLM-based mobile agents perform in terms of reasoning accuracy, execution accuracy, and reasoning-execution gaps across public benchmarks?
- **RQ3:** What is the impact of parameter scaling on reasoning accuracy, execution accuracy, and the magnitude of reasoning-execution gaps?

### 4.1 EXPERIMENT SETUP

We structure our experiments on three series of mobile-use agents and three challenging datasets to evaluate their reasoning accuracy (GTA) and execution accuracy (EM).

**Models**   For our experiments, we select three state-of-the-art mobile-use agents that represent the strongest open-source baselines across different benchmarks. **AgentCPM-GUI** (Zhang et al., 2025f) achieves leading performance on Chinese Android applications through cross-lingual training, progressive fine-tuning, and efficient on-device execution. **UI-TARS** (Qin et al., 2025) adopts a purely vision-driven, end-to-end architecture that surpasses even closed-source models like GPT-4o, supported by large-scale action datasets and self-iterative training. **GUI-Owl** (Ye et al., 2025), which serves as the backbone model for the Mobile-Agent-v3 (Ye et al., 2025) framework, establishes new state-of-the-art results across both desktop and mobile environments by integrating scalable environment infrastructure, self-evolving trajectory production, and reinforcement learning. All models are deployed following their official guide, detailed descriptions are provided in the Appendix C.

**Datasets**   We evaluate the above models on three benchmarks: AITZ (Zhang et al., 2024b), CAGUI (Zhang et al., 2025f), and AndroidControl (Li et al., 2024), where we adopt the high-level instruction split of AndroidControl, as it is more suitable for evaluating CoT reasoning quality. For each dataset, we collect the model-generated CoTs and predicted actions as evaluation data. To assess the reliability of our automated evaluation (RQ1), we apply stratified sampling across models and datasets to construct a subset of 1,800 instances for human annotation and agreement analysis. Our sampling procedure preserves the overall distribution of actions while also ensuring that representative minority cases are adequately covered. Figure 2 illustrates the action distribution of the original datasets and the manually sampled subset, showing that the sampling preserves the overall distributional characteristics. Detailed descriptions of the datasets are provided in the Appendix A.

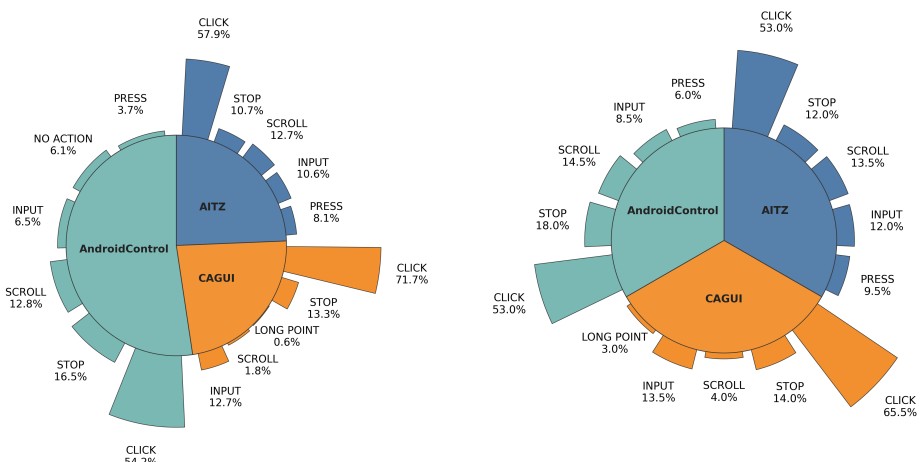

Figure 2: Action distributions of the original datasets and the stratified sampled subset. Overall, our sampling procedure preserves the overall distribution of actions while also ensuring that representative minority cases are adequately covered. Left shows the full dataset distributions, while right illustrates the 1,800 sampled instances used for human annotation and agreement analysis.

**GTA Evaluator**   For the implementation of $f(c_n)$, we adopt AgentCPM-GUI-8B (Zhang et al., 2025f) as the instruction-following VLM. We choose AgentCPM-GUI-8B (Zhang et al., 2025f) because it is an open-source multimodal agent trained on large-scale GUI trajectories and paired reasoning data, which makes it particularly suitable for grounding free-form CoT into executable actions. During evaluation, we apply deterministic decoding to obtain $f(c_n)$ for each reasoning trace and compare the predicted actions against the ground-truth labels under the same strict matching rule as EM. This setup allows us to fairly assess the consistency between model reasoning and execution across different benchmarks.

### 4.2 EVALUATOR RELIABILITY (RQ1)

Since the evaluator is designed to automatically map free-form CoTs into their implied actions and assess their correctness, it is crucial to examine whether such automatic judgments align with human annotations. Establishing the reliability of the evaluator ensures that subsequent large-scale evaluations and comparative studies rest on a solid methodological foundation. To assess reliability, we

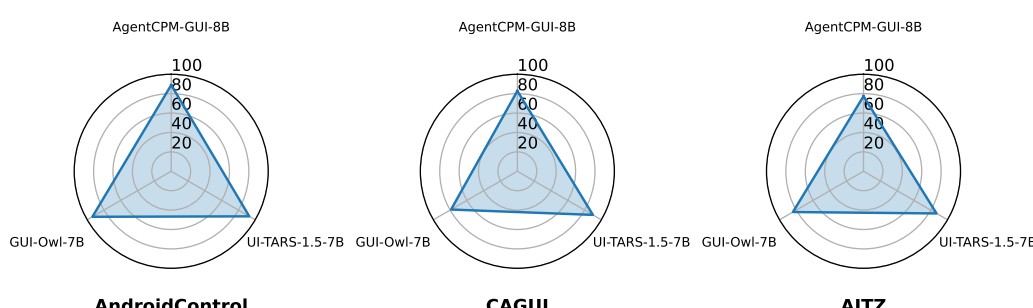

Figure 3: Radar plots show the **GTA evaluator accuracy** across three models and datasets. Overall, the evaluator achieves consistently high accuracy, with similar performance across models. Accuracy peaks on AndroidControl, while results on CAGUI and AITZ are slightly lower.

first construct a human-annotated reference set by sampling CoTs and labeling their GTA outcomes following the procedure described below.

Annotators are presented with the interface screenshot (with the ground-truth click-type action highlighted), the natural language instruction, the model-generated CoT, and the textual description of the ground-truth action. Given these inputs, annotators directly determine the GTA label according to the following criteria: (1) GTA = 1 if the CoT correctly implies the ground-truth action; (2) GTA = 0 if the CoT implies a different or incorrect action; (3) GTA = NA if the case is undecidable, typically due to erroneous ground-truth annotations or severely incomplete CoTs.

Each sample is independently annotated by two human experts. We retain only those instances where both annotators reach consensus on GTA $\in \{0, 1\}$. Samples for which at least one annotator assigns NA, or where the two annotators disagree, are discarded. This procedure ensures that the resulting human-annotated subset maintains high reliability and provides a solid reference for evaluating the automatic GTA evaluator.

We then apply the GTA Evaluator to the same samples and compute its prediction accuracy with respect to human consensus labels. This approach allows us to directly measure the degree to which the automatic evaluator replicates human judgment.

Radar plots in Figure 3 illustrate the accuracy of the GTA Evaluator across three models and three datasets. Overall, the evaluator achieves consistently high accuracy, with only minor variation across different models. Among the datasets, accuracy peaks on AndroidControl (Li et al., 2024), while performance on CAGUI (Zhang et al., 2025f) and AITZ (Zhang et al., 2024b) is slightly lower. These results suggest that the evaluator provides a stable and trustworthy proxy for human assessment.

The analysis indicates that the GTA Evaluator generalizes well across both models and datasets, reducing the need for costly and time-consuming human annotations. While minor dataset-specific differences remain, the overall robustness of the evaluator highlights its potential as a scalable evaluation protocol for future research. Beyond the current study, these findings point toward broader applications in automated assessment of reasoning-action alignment, where reliable and reproducible metrics are essential.

### 4.3 MODEL PERFORMANCE ASSESSMENT (RQ2)

Execution Match (EM) and Ground Truth Alignment (GTA) capture two complementary perspectives of model performance: EM measures strict execution accuracy, while GTA reflects whether the reasoning consistently implies the correct action. Evaluating them jointly is crucial, since high EM does not necessarily imply high GTA, and vice versa. We therefore examine EM and GTA across three benchmarks to assess both execution reliability and reasoning alignment.

We report EM and GTA scores for six representative models on AITZ (Zhang et al., 2024b), CAGUI (Zhang et al., 2025f), and AndroidControl (Li et al., 2024), where we adopt the high-level instruction split of AndroidControl, as it is more suitable for evaluating CoT reasoning quality. Results are visualized with grouped bar charts, where each dataset provides paired comparisons of EM

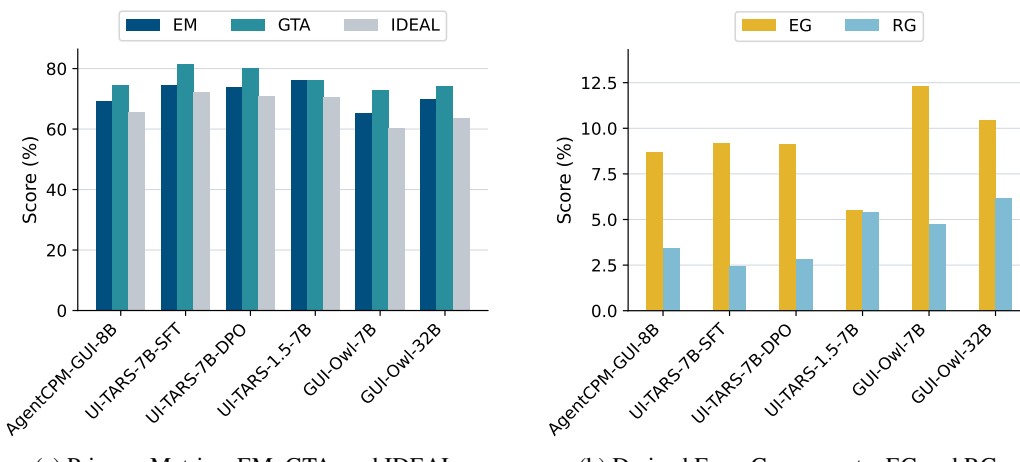

(a) Primary Metrics: EM, GTA, and IDEAL  (b) Derived Error Components: EG and RG

Figure 4: Model performance on the AndroidControl dataset. Execution accuracy is quantified by EM, whereas reasoning accuracy is measured by GTA. IDEAL represents the accuracy achievable under perfect reasoning and execution. We further decompose the performance gap using EG = GTA − IDEAL (Execution Gap) and RG = EM − IDEAL (Reasoning Gap). Cases where EG > RG suggest that execution, rather than reasoning, constitutes the primary performance bottleneck.

and GTA. This setup allows us to jointly interpret execution accuracy and reasoning accuracy, and to further inspect their divergence.

Overall, EM and GTA exhibit similar patterns across benchmarks, but important divergences emerge. From Figure 4, we have the following key findings.

**Bottleneck lies in EG.** Across 18 model-dataset combinations, GTA surpasses EM in 14 cases, indicating that EG > RG in the majority of scenarios. This suggests that most models can *reason correctly*, at least at the level of generating valid chains of thought, but encounter difficulty when mapping these intermediate reasoning steps into the precise executable actions required by the task. In other words, the primary bottleneck lies not in reasoning correctness itself, but in the reasoning-to-execution transition, where small misalignments or ambiguities in action formulation often lead to task failure. Such results highlight the importance of improving models ability to faithfully ground abstract reasoning in concrete actions, an ability that becomes especially critical in domains requiring fine-grained decision-making.

**Causal CoT Models often exhibit RG > EG.** We also observe counterexamples where EM exceeds GTA (e.g., AITZ results of AgentCPM and UI-TARS). In these cases, models rely heavily on action shortcuts learned during supervised fine-tuning, while ignoring or even contradicting their CoT reasoning. This phenomenon is particularly pronounced on AITZ, whose long and sometimes inconsistent CoTs amplify the problem: if the models reasoning ability is almost entirely inherited from SFT, it tends to overfit to actions and disregard CoT consistency.

**OOD data highlight different bottlenecks.** On the out-of-distribution benchmark CAGUI, untrained models consistently show high GTA but very large EG, suggesting that the main challenge lies in *grounding reasoning into unfamiliar screens*. Interestingly, the AgentCPM-GUI model, although achieving the best EM and GTA on CAGUI, still exhibits a large RG, indicating that its training on Chinese GUI data has encouraged reliance on action shortcuts. This observation highlights that stronger training can improve execution performance but may simultaneously exacerbate reasoning-faithfulness issues.

Representative qualitative examples for all four quadrants are provided in Appendix G.

## 4.4 PARAMETER SCALING EFFECTS (RQ3)

A central question in multimodal agent research is whether increasing model scale can effectively alleviate the reasoning-execution gap. While larger parameter counts typically enhance general reasoning abilities of large language models, it remains unclear whether such improvements translate to consistent gains in GUI interaction tasks, particularly in reducing execution errors.

We systematically evaluate UI-Tars (Qin et al., 2025) models of varying sizes (2B, 7B, and 72B) under two training paradigms (*SFT* and *DPO*) on the AndroidControl benchmark. We analyze their performance using four metrics: Execution Match (EM) and Ground-Truth Alignment (GTA) as positive indicators, and Execution Gap (EG) and Reasoning Gap (RG) as negative indicators. Visualization is provided in Figure 5, where point size encodes parameter scale.

As shown in Figure 5, scaling consistently improves both EM and GTA, with larger models achieving better alignment between predicted and ground-truth actions. At the same time, EG and RG decrease monotonically with parameter growth, indicating that scaling narrows the reasoning–execution gap. Notably, however, the largest 72B models still exhibit residual execution gaps (>10%), suggesting that scaling alone cannot fully eliminate misalignment. We also observe comparable trends across training paradigms, though DPO models achieve slightly stronger alignment than SFT models at the same scale.

These findings confirm that parameter scaling enhances both reasoning quality and execution accuracy in multimodal GUI agents, yet highlight diminishing returns in closing the reasoning-execution-gaps. This suggests that future work must move beyond pure scaling to incorporate training strategies or architectural designs that directly target reasoning-action consistency.

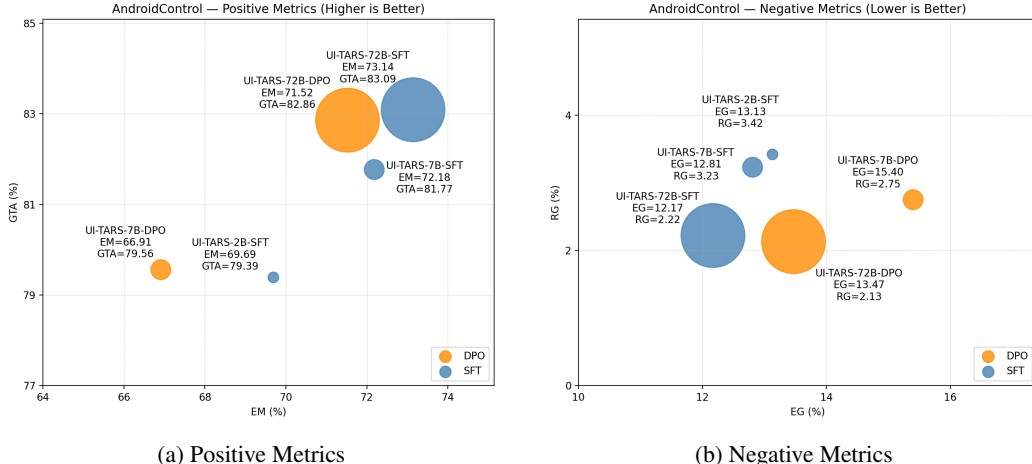

(a) Positive Metrics          (b) Negative Metrics

Figure 5: Effect of parameter scaling on reasoning–execution gaps in AndroidControl. (a) Positive metrics: EM and GTA, where higher is better. (b) Negative metrics: EG and RG, where lower is better. Orange points denote DPO models and blue points denote SFT models, with point size proportional to parameter scale. Scaling consistently improves EM and GTA while reducing EG and RG, though even the largest (72B) model still exhibits execution gaps above 10%.

## 5 CONCLUSION

In this work, we analyzed the reasoning-execution gaps in VLM-powered mobile-use agents. We introduced a framework to disentangle reasoning from execution in VLM-powered mobile-use agents. In contrast to prior work emphasizing action accuracy, our Ground-Truth Alignment (GTA) metric exposes reasoningexecution gaps through a four-quadrant diagnostic that highlights overlooked failure modes. Evaluations on three mobile benchmarks show that these gaps are common, with execution gaps dominating even in strong models. While scaling reduces misalignment, execution gaps remain, suggesting that progress cannot rely on size alone. By shifting evaluation from execution accuracy to reasoningexecution gaps, this work offers a step toward more transparent and trustworthy assessments of mobile-use agents.

ETHICS STATEMENT

This work studies reasoningexecution gaps in VLM-powered mobile-use agents. The models evaluated in this work were obtained from publicly available official repositories and used strictly under the licenses and usage terms specified by the original authors. All datasets employed in our experiments are publicly available and cited appropriately. For the small portion of data requiring human annotation, annotators were instructed to avoid recording any personal or sensitive information, and only samples with full agreement between annotators were retained to ensure reliability. Proper citations are included throughout the paper to ensure attribution and transparency.

REPRODUCIBILITY

To facilitate reproducibility, we provide detailed descriptions of model deployment and evaluation procedures in the Appendix C and Appendix D. During the submission phase, we have uploaded supplementary code with all author-identifying information anonymized. The supplementary materials include instructions for setting up environments, running the GTA evaluator, and replicating our experiments across the three benchmarks. Upon acceptance, we will publicly release the full codebase and scripts to ensure transparency and enable independent verification of our findings.

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

# A   APPENDIX: DATASET STATISTICS AND SAMPLING STRATEGY

In this appendix, we provide detailed statistics of the three evaluation datasets and describe the stratified sampling procedure used to construct the subset for human annotation.

## A.1   ACTION TYPE DISTRIBUTIONS

Table 1 summarizes the full action type distributions across AITZ, CAGUI, and AndroidControl. As can be seen, CLICK and SCROLL dominate most datasets, while other types such as PRESS or LONG POINT appear much less frequently.

| Action Type | AITZ | | CAGUI | | AndroidControl | |
|---|---|---|---|---|---|---|
| | Count | Ratio | Count | Ratio | Count | Ratio |
| CLICK | 2736 | 57.92% | 3237 | 71.68% | 5504 | 54.17% |
| STOP | 504 | 10.67% | 600 | 13.29% | 1680 | 16.53% |
| SCROLL | 601 | 12.72% | 79 | 1.75% | 1297 | 12.76% |
| INPUT | 500 | 10.58% | 574 | 12.71% | 685 | 6.74% |
| NO_ACTION | 0 | 0.00% | 1 | 0.02% | 623 | 6.13% |
| PRESS | 383 | 8.11% | 0 | 0.00% | 372 | 3.66% |
| LONG POINT | 0 | 0.00% | 25 | 0.55% | 0 | 0.00% |
| **Total** | 4724 | 100% | 4516 | 100% | 10161 | 100% |

Table 1: Full action type distribution across the three datasets.

To obtain a balanced yet representative evaluation set, we applied a stratified sampling scheme. Table 2 shows the resulting distribution after sampling 200 instances from each dataset while preserving diversity across action types.

| Action Type | AITZ | | CAGUI | | AndroidControl | |
|---|---|---|---|---|---|---|
| | Count | Ratio | Count | Ratio | Count | Ratio |
| CLICK | 106 | 53.00% | 131 | 65.50% | 106 | 53.00% |
| STOP | 24 | 12.00% | 28 | 14.00% | 36 | 18.00% |
| SCROLL | 27 | 13.50% | 8 | 4.00% | 29 | 14.50% |
| INPUT | 24 | 12.00% | 27 | 13.50% | 17 | 8.50% |
| PRESS | 19 | 9.50% | 0 | 0.00% | 12 | 6.00% |
| LONG POINT | 0 | 0.00% | 6 | 3.00% | 0 | 0.00% |
| **Total** | 200 | 100% | 200 | 100% | 200 | 100% |

Table 2: Action type distribution after stratified sampling (200 samples per dataset).

## A.2   SAMPLING METHOD

We adopted a stratified sampling procedure with minimum allocation and paired projection. For each dataset with stratum counts $\{n_c\}$ and target size $N$, we first allocate

$$m_c = \min(k, n_c), \quad M = \sum_c m_c, \quad R = N - M,$$

then distribute the remainder proportionally using the largest remainder method:

$$t_c = \min\left(n_c, m_c + \lfloor R \cdot n_c / \sum_j n_j \rfloor + \delta_c\right), \qquad \sum_c t_c = N.$$

On a baseline model, we sample per stratum to form a key list `key_list` = $\{(\texttt{episode\_id}, \texttt{step\_id})\}$, and then project the same keys onto other models to guarantee alignment.

Code Listing 1: Stratified sampling with paired projection

```
# Input: counts {n_c}, total N, minimum k, dataset D
m_c = min(k, n_c)          # per stratum minimum
M = sum(m_c for c in C)
R = N − M
q_c = R ∗ n_c / sum(n_j)    # proportional share
a_c = floor(q_c)
L = R − sum(a_c)
delta_c = distribute_L_by_largest_remainder(q_c − a_c)
t_c = min(n_c, m_c + a_c + delta_c)

# Draw without replacement to build key_list
key_list = sample_per_stratum(D, t_c)
# For other models, filter by key_list
```

## B  APPENDIX: HUMAN ANNOTATION

## C  APPENDIX: MODELS DEPLOYMENT

**AgentCPM-GUI (Zhang et al., 2025f).** This model family adopts a single-turn format without history:

$$P(c_n, a_n \mid I, o_n), \tag{9}$$

which is equivalent to setting $H_n = \varnothing$.

---

**AgentCPM-GUI Data Examples**

**System Prompt**

# Role
You are an intelligent agent familiar with Android touchscreen GUI operations.
Based on the user's query, you will analyze the GUI elements and layout of the current
    interface, and generate the corresponding operation.

# Task
Given the user's query and the current screen screenshot, output the next step operation.

# Rule
− Output must be in compact JSON format.
− The operation must follow the Schema constraints.

# Schema
```json
{
 "type": "object",
 "description": "Execute an operation and decide the current task status",
 "additionalProperties": false,
 "properties": {
  "thought": {
   "type": "string",
   "description": "Reasoning process of the agent"
  },
  "POINT": {
   "$ref": "#/$defs/Location",
   "description": "Tap at the specified position on the screen"
  },
```

---

```
      "to": {
       "description": "Movement, combined gesture parameters",
       "oneOf": [
         {
          "enum": [ "up", "down", "left", "right" ],
          "description": "From the current point (POINT), perform a swipe gesture in one of
         the four directions"
         },
         {
          "$ref": "#/$defs/Location",
          "description": "Move to a specific position"
         }
       ]
      },
      "duration": {
       "type": "integer",
       "description": "Execution time or waiting time in milliseconds",
       "minimum": 0,
       "default": 200
      },
      "PRESS": {
       "type": "string",
       "description": "Trigger special keys. HOME = go to home screen, BACK = back button
         , ENTER = enter key",
       "enum": [ "HOME", "BACK", "ENTER" ]
      },
      "TYPE": {
       "type": "string",
       "description": "Input text"
      },
      "STATUS": {
       "type": "string",
       "description": "Current task status. Special cases: satisfied = no action required;
         impossible = task cannot be completed; interrupt = task interrupted; need_feedback =
         user feedback required",
       "enum": [ "continue", "finish", "satisfied", "impossible", "interrupt", "need_feedback"
         ],
       "default": "continue"
      }
     },
     "$defs": {
      "Location": {
       "type": "array",
       "description": "Coordinates relative to the top−left corner of the screen, scaled between
         0−1000 by width and height. First element = x, second element = y",
       "items": { "type": "integer", "minimum": 0, "maximum": 1000 },
       "minItems": 2,
       "maxItems": 2
      }
     }
    }
```

**User**

<Question>[query]</Question>\nCurrent screen screenshot:
[current_screenshot]

---

**Assistant**

[thought_and_action]

---

**UI-TARS (Qin et al., 2025).** This model family organizes the task as a multi-turn dialogue, retaining at most the last $N$ interaction triples in the context. Specifically,

$$H_n = \{(o_j, c_j, a_j)\}_{j=\max(1,\, n-N)}^{n-1}, \tag{10}$$

where $N = 4$ in our setting. The model predicts

$$P(c_n, a_n \mid I, H_n, o_n). \tag{11}$$

---

**UI-TARS Data Example**

**System Message**

You are a helpful assistant.

**User**

You are a GUI agent. You are given a task and your action history, with screenshots. You
     need to perform the next action to complete the task.
## Output Format
```
Thought: ...
Action: ...
```
## Action Space

click(point='<point>x1 y1</point>')
long_press(point='<point>x1 y1</point>')
type(content='') #If you want to submit your input, use "\\n" at the end of 'content'.
scroll(point='<point>x1 y1</point>', direction='down or up or right or left')
press_home()
press_back()
finished(content='xxx') # Use escape characters \\', \\", and \\n in content part to ensure
     we can parse the content in normal python string format.

## Note
− Use {language} in 'Thought' part.
− Write a small plan and finally summarize your next action (with its target element) in one
     sentence in 'Thought' part.

## User Instruction
{instruction}

**User**

[history_screenshot]

**Assistant**

[history_thought_and_action]

**User**

[current_screenshot]

---

---

**Assistant**

[thought_and_action]

---

**GUI-Owl (Ye et al., 2025).** This model family also uses a single-turn format but supplements the input with a compressed history representation $\tilde{H}_n$:

$$P(c_n, a_n \mid I, \tilde{H}_n, o_n), \tag{12}$$

where $\tilde{H}_n = \mathrm{Compress}(H_n)$ denotes a textual summary of past interactions.

---

**GUI-Owl Data Example**

**System Message**

You are a helpful assistant.

# Tools

You may call one or more functions to assist with the user query.

You are provided with function signatures within <tools></tools> XML tags:
<tools>
{"type": "function", "function": {"name_for_human": "mobile\_use", "name": "mobile\
    _use", "description": "Use a touchscreen to interact with a mobile device, and take
    screenshots.
∗ This is an interface to a mobile device with touchscreen. You can perform actions like
    clicking, typing, swiping, etc.
∗ Some applications may take time to start or process actions, so you may need to wait and
    take successive screenshots to see the results of your actions.
∗ The screen's resolution is {width}x{height}.
∗ Make sure to click any buttons, links, icons, etc with the cursor tip in the center of the
    element. Don't click boxes on their edges unless asked.", "parameters": {"properties":
    {"action": {"description": "The action to perform. The available actions are:
∗ 'key': Perform a key event on the mobile device.
    − This supports adb's 'keyevent' syntax.
    − Examples: \"volume\_up\", \"volume\_down\", \"power\", \"camera\", \"clear\".
∗ 'click': Click the point on the screen with coordinate (x, y).
∗ 'long\_press': Press the point on the screen with coordinate (x, y) for specified seconds.
∗ 'swipe': Swipe from the starting point with coordinate (x, y) to the end point with
    coordinates2 (x2, y2).
∗ 'type': Input the specified text into the activated input box.
∗ 'system\_button': Press the system button.
∗ 'open': Open an app on the device.
∗ 'wait': Wait specified seconds for the change to happen.
∗ 'terminate': Terminate the current task and report its completion status.", "enum": ["key
    ", "click", "long\_press", "swipe", "type", "system\_button", "open", "wait", "
    terminate"], "type": "string"}, "coordinate": {"description": "(x, y): The x (pixels
    from the left edge) and y (pixels from the top edge) coordinates to move the mouse to.
     Required only by 'action=click', 'action=long\_press', and 'action=swipe'.", "type":
    "array"}, "coordinate2": {"description": "(x, y): The x (pixels from the left edge) and
    y (pixels from the top edge) coordinates to move the mouse to. Required only by '
    action=swipe'.", "type": "array"}, "text": {"description": "Required only by 'action=
    key', 'action=type', and 'action=open'.", "type": "string"}, "time": {"description": "
    The seconds to wait. Required only by 'action=long\_press' and 'action=wait'.", "
    type": "number"}, "button": {"description": "Back means returning to the previous
    interface, Home means returning to the desktop, Menu means opening the application

```
background menu, and Enter means pressing the enter. Required only by 'action=
system\_button'", "enum": ["Back", "Home", "Menu", "Enter"], "type": "string"}, "
status": {"description": "The status of the task. Required only by 'action=terminate'.",
 "type": "string", "enum": ["success", "failure"]}}, "required": ["action"], "type": "
object"}, "args\_format": "Format the arguments as a JSON object."}}
</tools>

For each function call, return a json object with function name and arguments within <tool
    \_call></tool\_call> XML tags:
<tool_call>
{"name": <function−name>, "arguments": <args−json−object>}
</tool_call>
```

**User**

The user query: [user_request]
Task progress (You have done the following operation on the current device): [
    history_actions]
Before answering, explain your reasoning step−by−step in <thinking></thinking> tags,
    and insert them before the <tool_call></tool_call> XML tags.
After answering, summarize your action in <conclusion></conclusion> tags, and insert
    them after the <tool_call></tool_call> XML tags.

[current_screenshot]

**Assistant**

[thought_and_action]

## D    APPENDIX: EVALUATION PROCEDURES

To ensure reproducibility of our results, we release supplementary code together with detailed instructions in the repository. The main evaluation procedures are as follows:

1. **Environment setup.** All experiments were conducted on Linux with Python 3.10/3.11 and CUDA-enabled GPUs. The repository provides a `requirements.txt` file to reproduce the exact software environment.

2. **Model preparation.** Pre-trained checkpoints of AgentCPM-GUI, UI-TARS, and GUI-Owl are downloaded following the official instructions (see `model/README.md`). Local paths are configured as required by the inference scripts.

3. **Dataset preparation.** Evaluation datasets (AITZ, CAGUI, AndroidControl) are publicly available. We provide preprocessing scripts and instructions in `eval/eval_data/README.md`, which reproduce the splits used in our experiments.

4. **Inference and EM evaluation.** For each modeldataset pair, we run inference scripts (`run_predict_*.py`) to generate action predictions, followed by `run_eval_agent.py` to compute Exact Match (EM) scores.

5. **GTA evaluation.** To evaluate reasoning correctness, we provide scripts to extract CoT traces and apply the GTA evaluator (`eval/README_COT.md`). The evaluation enforces strict type and parameter matching consistent with the EM metric.

6. **Manual annotation (optional).** For RQ1, we use a lightweight annotation interface (`cot_eval/annot_ui_min.py`) to construct a human-annotated reference set. Disagreements are filtered, and only consensus labels are retained.

7. **Analysis and visualization.** Quadrant analysis and diagnostic plots are generated with scripts in `cot_eval/`, reproducing all figures reported in the main text.

This procedure allows independent researchers to replicate all experiments reported in the paper, from model inference to reasoningexecution gap analysis.

## E    APPENDIX: ADDITIONAL RESULTS

In this appendix, we provide supplementary quantitative results to complement the main text. Table 3 reports the reliability of the GTA Evaluator, showing that it achieves consistently high accuracy across datasets and models. Table 4 presents benchmark results on AITZ, CAGUI, and AndroidControl, offering a more detailed comparison of reasoning and execution metrics (EM, GTA, EG, and RG). Together, these results further validate the robustness of our evaluation framework and highlight distinct performance patterns across model families and datasets.

| Dataset | AgentCPM-GUI-8B | | | UI-TARS-1.5-7B | | | GUI-Owl-7B | | |
|---|---|---|---|---|---|---|---|---|---|
| | AITZ | CAGUI | AC | AITZ | CAGUI | AC | AITZ | CAGUI | AC |
| Valid samples | 159 | 169 | 135 | 153 | 168 | 155 | 150 | 180 | 162 |
| GTA Accuracy (%) | 77.36 | 82.84 | 88.89 | 86.93 | 89.88 | 92.90 | 84.00 | 78.89 | 93.83 |

Table 3: Reliability of the GTA Evaluator. Overall, the evaluator achieves consistently high accuracy, with similar performance across models. Accuracy peaks on AndroidControl (AC), while results on CAGUI and AITZ are slightly lower.

| Model | AITZ (%) | | | | CAGUI (%) | | | | AndroidControl (%) | | | |
|---|---|---|---|---|---|---|---|---|---|---|---|---|
| | EM | GTA | EG | RG | EM | GTA | EG | RG | EM | GTA | EG | RG |
| AgentCPM-GUI-8B | **76.29** | **70.43** | **3.51** | 9.38 | **91.21** | **85.35** | **2.28** | 8.13 | 69.17 | 74.45 | 8.68 | 3.41 |
| UI-TARS-7B-SFT | 66.69 | 64.98 | 7.73 | 9.44 | 71.36 | 77.86 | 9.08 | 2.57 | 74.71 | **81.44** | 9.18 | **2.45** |
| UI-TARS-7B-DPO | 65.95 | 63.86 | 8.48 | 10.57 | 70.60 | 78.52 | 10.35 | 2.44 | 73.87 | 80.16 | 9.15 | 2.85 |
| UI-TARS-1.5-7B | 58.94 | 66.09 | 11.85 | **4.69** | 68.42 | 81.68 | 15.32 | **2.06** | **76.10** | 76.22 | **5.53** | 5.41 |
| GUI-Owl-7B | 61.01 | 66.17 | 10.29 | 5.12 | 61.00 | 79.87 | 21.75 | 2.88 | 65.22 | 72.79 | 12.31 | 4.74 |
| GUI-Owl-32B | 59.65 | 66.09 | 12.19 | 5.76 | 65.88 | 81.53 | 18.29 | 2.64 | 69.89 | 74.15 | 10.43 | 6.18 |

Table 4: Benchmark results on AITZ, CAGUI, and AndroidControl datasets. Best results are marked in bold face. Overall, AgentCPM-GUI shows leading performance on AITZ and CAGUI, while UI-TARS achieves the strongest results on AndroidControl.

## F    APPENDIX: THE USE OF LARGE LANGUAGE MODELS

In preparing this paper, we used ChatGPT 5 to aid or polish the writing, particularly for grammar and stylistic refinement. The authors take full responsibility for all the content presented under their names, including any parts where LLMs were involved. The use of LLMs did not affect the design, execution, or interpretation of the research itself.

## G    APPENDIX: CASE STUDY

In the following case studies, we visualize four representative outcomes of reasoningexecution alignment using **UI-TARS-1.5-7B**. Each figure shows the mobile screenshot annotated with two markers: green dots denote the ground-truth click locations, while blue dots indicate the models predicted clicks. The examples cover all four quadrants of our diagnostic framework: Both Right (Ideal), Both Wrong, Execution Gap (EG), and Reasoning Gap (RG).

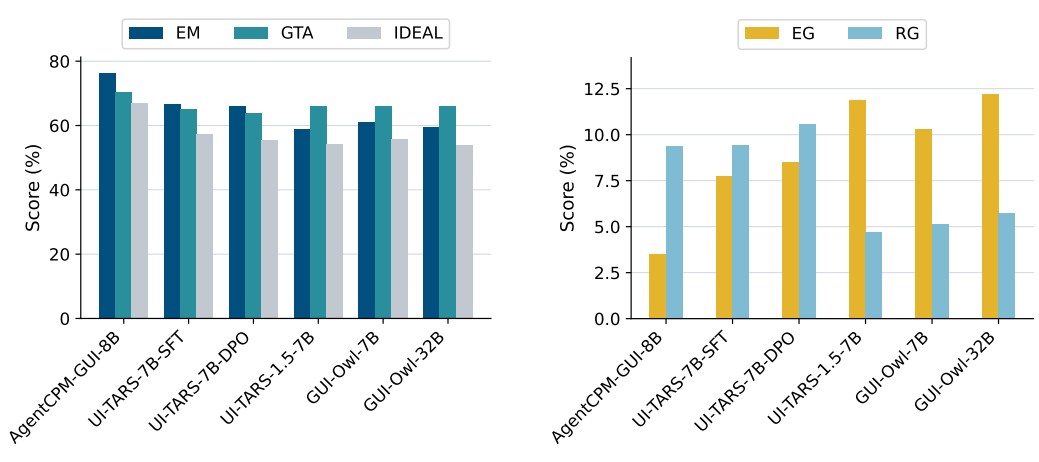

(a) Primary Metrics: EM, GTA, and IDEAL

(b) Derived Error Components: EG and RG

Figure 6: Model performance on the AITZ dataset. Execution accuracy is quantified by EM, whereas reasoning accuracy is measured by GTA. IDEAL represents the accuracy achievable under perfect reasoning and execution. We further decompose the performance gap using EG = GTA − IDEAL (Execution Gap) and RG = EM − IDEAL (Reasoning Gap). Cases where EG > RG suggest that execution, rather than reasoning, constitutes the primary performance bottleneck.

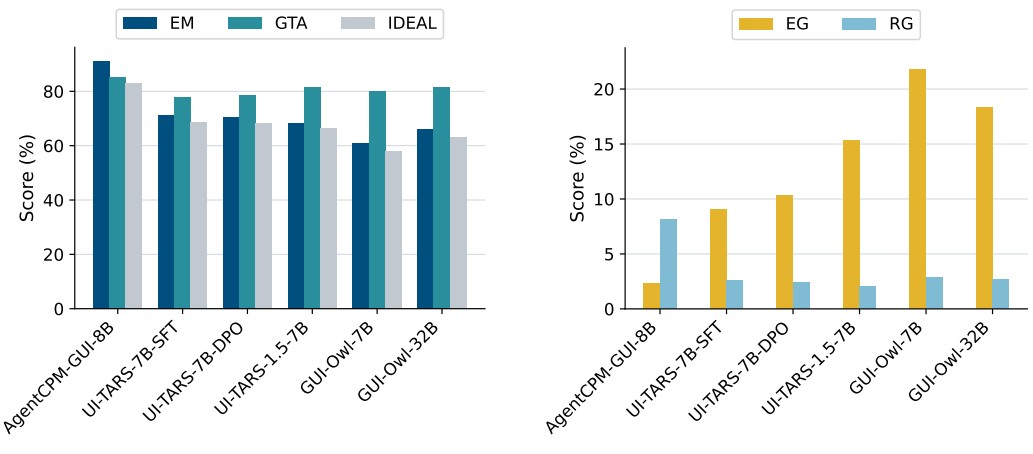

(a) Primary Metrics: EM, GTA, and IDEAL

(b) Derived Error Components: EG and RG

Figure 7: Model performance on the CAGUI dataset. Execution accuracy is quantified by EM, whereas reasoning accuracy is measured by GTA. IDEAL represents the accuracy achievable under perfect reasoning and execution. We further decompose the performance gap using EG = GTA − IDEAL (Execution Gap) and RG = EM − IDEAL (Reasoning Gap). Cases where EG > RG suggest that execution, rather than reasoning, constitutes the primary performance bottleneck.

1188
1189
1190
1191
1192
1193
1194
1195
1196
1197
1198
1199
1200
1201
1202
1203
1204
1205
1206
1207
1208
1209
1210
1211
1212
1213
1214
1215
1216
1217
1218
1219
1220
1221
1222
1223
1224
1225
1226
1227
1228
1229
1230
1231
1232
1233
1234
1235
1236
1237
1238
1239
1240
1241

**Case: Both Right (Ideal)**

**Instruction:** Add Lenovo ThinkPad to the cart on Target.

**CoT:** I noticed that the search results page has loaded, and the first item displayed is the Lenovo ThinkPad T570 laptop. To proceed with the task of adding it to the cart, I need to click on the product image or title to access the product details page. This will allow me to see more information about the product and the option to add it to the cart.

**Prediction:** CLICK(1259, 414)

**Ground Truth:** CLICK(1262, 464)

**Scores:** EM = 1, GTA = 1

Figure 8: Case study of a Both Right (Ideal) example.

**Case: Both Wrong**

**Instruction:** Check the best rated table lamp on Crate and Barrel.

**CoT:** I noticed that there's a pop-up at the bottom of the page offering a 10% discount, which is going to interfere with my ability to browse the products. I need to close it first, so I'll click the close button in the upper right corner. This way, I can focus on finding the best-rated table lamps.

**Prediction:** CLICK(864, 488)

**Ground Truth:** CLICK(596, 56)

**Scores:** EM = 0, GTA = 0

Figure 9: Case study of a Both Wrong example.

**Case: Execution Gap (EG)**

**Instruction:** Go to internet settings.

**CoT:** I noticed that the current page is the "Hotspot & tethering" settings interface, which is not the internet settings we are looking for. Let me click the back button in the upper left corner to return to the previous menu and continue searching for the correct settings option.

**Prediction:** STOP (terminate task)

**Ground Truth:** CLICK(82, 38)

**Scores:** EM = 0, GTA = 1

Figure 10: Case study of an Execution Gap example.

**Case: Reasoning Gap (RG)**

**Instruction:** Since I am traveling, I would like to listen to news podcasts on the New York Times app.

**CoT:** I'm currently on the health section page of the New York Times app, and it seems that this isn't the right place to find podcasts. I need to go back to the homepage to look for the podcasts section. Let me click the back button in the top left corner to return to the main page and continue my search.

**Prediction:** PRESS(back)

**Ground Truth:** PRESS(back)

**Scores:** EM = 1, GTA = 0

Figure 11: Case study of a Reasoning Gap example.

**Case: Reasoning Gap (RG)**

**Instruction:** Open the pinterest app and create an art on the photo from the gallery then publish the artwork.

**CoT:** I have completed all the steps for creating and editing the artwork, and now I see that the final step is to share it. I need to click on the share icon located at the bottom right corner of the screen to make this piece of art available to others.

**Prediction:** STOP

**Ground Truth:** STOP

**Scores:** EM = 1, GTA = 0

Figure 12: Case study of a Reasoning Gap example 2.

**Evaluator Success Case**

**Instruction:** Toggle notifications for the Google Maps app

**CoT:** The "Settings" option visible on the screen is relevant to the given query as it is likely to lead to notification settings for Google Maps. Possible actions are to click on the "Settings" option to proceed with toggling notifications. click on the Settings located at the lower left part of the screen.

**Evaluator Prediction:** CLICK (733, 58)

**Ground Truth:** CLICK (732, 42)

**Scores:** $\text{GTA}_{\text{human}} = 1$, $\text{GTA}_{\text{evaluator}} = 1$

Figure 13: Evaluator Success Case.

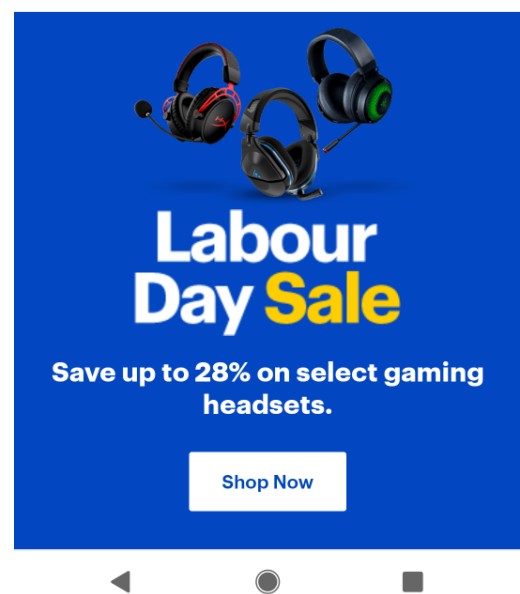

**Evaluator Failure Case**

**Instruction:** Look up the best gaming headphones on Best Buy

**CoT:** The screen shows a sale on gaming headsets, but it does not provide information on the best options available. Possible actions are to browse through the selection of headsets listed under the sale, use the search function within the page to find specific products, or navigate to other sections of the site that might offer more detailed product comparisons and reviews. stop and set the query as completed

**Evaluator Prediction:** `Scroll Down`

**Ground Truth:** `STOP`

**Scores:** $\text{GTA}_{\text{human}} = 1$, $\text{GTA}_{\text{evaluator}} = 0$

Figure 14: Evaluator Failure Case.

