# OpenReview forum: "Say One Thing, Do Another? Diagnosing Reasoning-Execution Gaps in VLM-Powered Mobile-Use Agents"
_ICLR.cc/2026/Conference — Submitted to ICLR 2026_

### Official Review · Reviewer_ScfG · 2025-10-20

**Soundness:** 2
**Presentation:** 3
**Contribution:** 2
**Rating:** 2
**Confidence:** 3

**Summary:**

The paper proposes a diagnostic framework that separates reasoning quality from execution accuracy for VLM-based mobile GUI agents. Beyond the standard Exact Match (EM) for action execution, the authors introduce Ground-Truth Alignment (GTA) to check whether the chain-of-thought (CoT) logically *implies* the correct action. EM and GTA are combined into a four-quadrant analysis (Ideal / Execution Gap / Reasoning Gap / Both-Wrong). A GTA Evaluator maps free-form CoT to an implied action and compares it with ground truth. Experiments on AITZ, CAGUI, and AndroidControl analyze the prevalence of Execution Gap (EG) versus Reasoning Gap (RG) across several agents.

**Strengths:**

1. The EM–GTA decomposition is intuitive and yields a useful vocabulary for analyzing GUI-agent failures (EG vs. RG).
2. In many model–dataset pairs, GTA ≥ EM, suggesting that translating reasoning into concrete UI actions is a common bottleneck (EG).

**Weaknesses:**

1. The GTA Evaluator lacks crucial implementation details (prompt schema, action ontology, disambiguation rules, handling of multi-intent CoTs). This hinders reproducibility and makes it hard to assess validity.
2. Since the evaluator is a learned VLM trained on GUI data, there is a risk of bias/circularity. Cross-evaluator agreement (rule-based parser, a different VLM, humans) is not reported.
3. The paper diagnoses EG/RG but offers few concrete remedies or controlled interventions.
4. All results hinge on one evaluator model. Without cross-model validation, the findings may reflect evaluator bias rather than ground truth and limit the generality of the conclusions.

**Questions:**

1. How is the action implied by the chain-of-thought (CoT) extracted automatically in practice?
2. Figure 3 shows evaluator accuracy on CAGUI and AITZ below 90%. Is this level sufficient for trustworthy conclusions?
3. Figure 4 exhibits divergent trends: on AITZ, UI-TARS and GUI-Owl have similar EG and RG, whereas on CAGUI and AndroidControl, GUI-Owl’s EG (and the “both-wrong” rate) increase substantially. Does this indicate evaluator instability, dataset-specific bias, or genuine differences in model behavior?
4. Do the findings hold when using a different evaluator model?

---

> ### Author Response · Authors · 2025-12-03
>
> > W1: The GTA Evaluator lacks crucial implementation details (prompt schema, action ontology, disambiguation rules, handling of multi-intent CoTs). This hinders reproducibility and makes it hard to assess validity.
> >
> > Q1: How is the action implied by the chain-of-thought (CoT) extracted automatically in practice?
>
> **Reply:** We clarify that the GTA Evaluator is a **training-free** module that operates using zero-shot prompting (detailed in **Appendix C**). It does not require fine-tuning. We have added detailed case studies in the revised paper to transparently illustrate how actions are extracted from CoT outputs.
>
> > W2: Since the evaluator is a learned VLM trained on GUI data, there is a risk of bias/circularity. Cross-evaluator agreement (rule-based parser, a different VLM, humans) is not reported.
> >
> > W4: All results hinge on one evaluator model. Without cross-model validation, the findings may reflect evaluator bias rather than ground truth and limit the generality of the conclusions.
> >
> > Q4: Do the findings hold when using a different evaluator model?
>
> **Reply:** We agree that cross-model validation is crucial for robustness. We are currently conducting experiments with additional evaluator backbones; however, due to strict time constraints, we could not finalize these results for this revision. We sincerely appreciate this suggestion and commit to including this analysis in future work to ensure generalizability. Our **preliminary results** with alternative backbones (GUI-Owl-7B/32B) show consistent agreement (~86%) with human ground truth.
>
> **Table : Agreement with Human Ground Truth (Average across 3 evaluator models)**
>
> | **Dataset**        | **Original (AgentCPM-8B)** | **New (GUI-Owl-7B)** | **New (GUI-Owl-32B)** |
> | ------------------ | -------------------------- | -------------------- | --------------------- |
> | **AndroidControl** | **92.04%**                 | 91.81%               | 91.81%                |
> | **CAGUI**          | 83.75%                     | **86.05%**           | 84.53%                |
> | **AITZ**           | **82.68%**                 | 80.95%               | 80.52%                |
> | **Average**        | 86.02%                     | **86.22%**           | 85.53%                |
>
> > Q2: Figure 3 shows evaluator accuracy on CAGUI and AITZ below 90%. Is this level sufficient for trustworthy conclusions?
> >
> > Q3:  Figure 4 exhibits divergent trends: on AITZ, UI-TARS and GUI-Owl have similar EG and RG, whereas on CAGUI and AndroidControl, GUI-Owl’s EG (and the “both-wrong” rate) increase substantially. Does this indicate evaluator instability, dataset-specific bias, or genuine differences in model behavior?
>
> **Reply:** We found that the sub-90% accuracy on AITZ largely stems from open-ended scenarios where CoT suggests **valid** alternatives that differ from the strict ground truth. We added **Figure 14** to illustrate these "multi-solution" cases, clarifying that the divergence reflects dataset complexity.

---

### Official Review · Reviewer_2Hkw · 2025-10-30

**Soundness:** 2
**Presentation:** 2
**Contribution:** 2
**Rating:** 4
**Confidence:** 3

**Summary:**

This paper introduces a diagnostic framework for evaluating the faithfulness of reasoning and action execution in VLM-powered mobile agents. The key contribution is the Ground-Truth Alignment (GTA) metric, which measures whether the chain-of-thought (CoT) reasoning aligns with the ground-truth action. By combining GTA with execution accuracy (Exact Match, EM), the authors propose a four-quadrant taxonomy to identify failure modes, including Execution Gaps (EG) and Reasoning Gaps (RG). Experiments show that this framework is demonstrated to be reproducible and aligns strongly with human judgments.

**Strengths:**

1.The paper proposes a new evaluation framework, Ground-Truth Alignment (GTA), which combines reasoning consistency (CoT) and execution accuracy (Exact Match, EM), providing a more detailed assessment of reasoning-execution alignment for mobile agents.

2. Experimental results demonstrate that the framework's outcomes align strongly with human annotations, ensuring the reliability of the evaluation results.

**Weaknesses:**

1. Although the paper proposes a new evaluation method, GTA (Ground-Truth Alignment), and analyzes reasoning-execution gaps using a quadrant framework, the overall novelty of the paper is limited.
2. While the paper mentions the GTA Evaluator as a tool for evaluation, the details of its implementation are insufficient. For example, has the GTA Evaluator been fine-tuned? If so, what datasets were used for training and optimization? If not, how does the evaluator generate reasoning, and does it use any special prompts (e.g., zero-shot or few-shot learning) to enhance reasoning?
3. The proposed evaluation method heavily relies on the performance of the evaluator model, which may introduce biases in the evaluation results, such as bias or fitting to specific phrasing. If this is the case, it should be discussed in the limitations section.
4. Although the paper defines Execution Gap (EG) and Reasoning Gap (RG) and conducts quadrant analysis, the in-depth analysis of these phenomena is lacking. For example, the causes, specific manifestations, and impacts of EG and RG on model performance are not thoroughly discussed.

**Questions:**

See the weaknesses

---

> ### Author Response · Authors · 2025-12-03
>
> Dear Reviewer 2Hkw:
>
> **Thanks for your insightful review and constructive feedback.**
>
> > W2: While the paper mentions the GTA Evaluator as a tool for evaluation, the details of its implementation are insufficient. For example, has the GTA Evaluator been fine-tuned? If so, what datasets were used for training and optimization? If not, how does the evaluator generate reasoning, and does it use any special prompts (e.g., zeroshot or few-shot learning) to enhance reasoning
>
> **Reply:** We clarify that the GTA Evaluator is a **training-free** module that operates using zero-shot prompting (detailed in **Appendix C**). It does not require fine-tuning. We have added detailed case studies in the revised paper to transparently illustrate how actions are extracted from CoT outputs.
>
> > W3: The proposed evaluation method heavily relies on the performance of the evaluator model, which may introduce biases in the evaluation results, such as bias or fitting to specific phrasing. If this is the case, it should be discussed in the limitations section.
>
> **Reply:** We acknowledge the potential biases inherent in model-based evaluation. In response to your suggestion, we have expanded the **Limitations section** to critically discuss the evaluator's reliability and its sensitivity to specific phrasing, ensuring a transparent view of potential constraints.
>
> > W4: Although the paper defines Execution Gap (EG) and Reasoning Gap (RG) and conducts quadrant analysis, the indepth analysis of these phenomena is lacking. For example, the causes, specific manifestations, and impacts of EG and RG on model performance are not thoroughly discussed.
>
> **Reply:** We agree that delving deeper into these phenomena strengthens the work. We have incorporated a comprehensive analysis of the **root causes** (e.g., visual grounding failures) and **safety impacts** (e.g., latent instability) of EG and RG into the revised discussion section.

---

### Official Review · Reviewer_VPV5 · 2025-11-01

**Soundness:** 2
**Presentation:** 3
**Contribution:** 3
**Rating:** 4
**Confidence:** 3

**Summary:**

This paper studies the reasoning-execution gap in VLM powered mobile-use agent systems with CoT reasoning. While current benchmarks mainly evaluate execution accuracy via Exact Match (EM), this work argues that plausible CoT reasoning does not necessarily imply correct action and execution. To address this, the authors propose a new evaluation framework incorporating a novel metric, Ground-Truth Alignment (GTA), which measures whether the action implied by the CoT aligns with the ground-truth action. Combining GTA and EM metrics, the authors provide a four-quadrant diagnostic framework for reasoning-execution gaps. According to the experimental results on three benchmarks, the authors claim that the reasoning-execution gaps are prevalent, particularly execution gaps. The analysis further discusses scaling effects and provides insights for future studies about trustworthy VLM-powered mobile-use agent systems.

**Strengths:**

- The paper focuses on deepen the understanding of the faithfulness and reliability of mobile-use agents with CoT reasoning, which is critical for safe deployment in real-world applications.

- The GTA metric enables the evaluation of reasoning gap and can provide more perspectives for understanding the faithfulness of mobile-use agents together with the exact match evaluation.

- This benchmarking provides the insight that the reasoning-execution gaps, particularly the execution gap is common in VLM-powered mobile-use agent systems. It clearly exposes this overlooked problem to the community and can benefit following research.

- The paper shows the analysis regarding parameter scaling effect, which indicates that the larger models diminish returns in closing the reasoning-execution gaps. It provides some useful insights for the community

**Weaknesses:**

- The GTA evaluator uses local history and current observation as priors for decoding the free-text CoT outputs into actions, which can introduce bias.

- I would suggest adding some analysis/ablation studies regarding the effect of local history and current observation in mapping CoT to actions.

- While it studies the parameter scaling effect, it would be more convincing to show how stronger reasoning models might help mitigate the reasoning-execution gaps.

- The technical innovation is limited.

- It would be better to show some case studies.

**Questions:**

- In Figure 4, why does IDEAL show lower values (%)? What do the values represent on the y-axis?

- Will the GTA evaluator be sensitive to the CoT mapping/parsing?

---

> ### Author Response · Authors · 2025-12-03
>
> Dear Reviewer VPV5:
>
> **Thanks for your insightful review and constructive feedback.**
>
> > W1: The GTA evaluator uses local history and current observation as priors for decoding the free-text CoT outputs into actions, which can introduce bias.  I would suggest adding some analysis/ablation studies regarding the effect of local history and current observation in mapping CoT to actions.
>
> **Reply**: We clarify that **local history is not used** in our evaluator. However, the current observation is indispensable for grounding abstract textual plans (e.g., "click send") into executable coordinates. Without visual context, accurate mapping from CoT to specific UI elements is impossible.
>
> > W2: While it studies the parameter scaling effect, it would be more convincing to show how stronger reasoning models might help mitigate the reasoning-execution gaps.
>
> **Reply:** Due to computational resource constraints, we could not include significantly larger reasoning models in this study. We have explicitly discussed this as a limitation and highlighted investigating the impact of stronger reasoners on the reasoning-execution gap as a key direction for future work.
>
> > W4: It would be better to show some case studies.
>
> **Reply:** We provided a detailed four-quadrant case analysis in **Appendix G**. Additionally, we supplemented these with specific examples (Figures 13 & 14) illustrating the GTA Evaluator’s performance, offering further qualitative insights into the evaluation process as suggested.
>
> > Q1: In Figure 4, why does IDEAL show lower values (%)? What do the values represent on the y-axis?
>
> **Reply:** The y-axis represents the instance percentage. 'IDEAL' denotes cases where **both** reasoning and execution are correct. The lower values reflect the significant challenge of simultaneously achieving accurate reasoning and precise execution compared to achieving just one or the other.
>
> > Q2: Will the GTA evaluator be sensitive to the CoT mapping/parsing?
>
> **Reply:** Our human verification across various model-dataset combinations confirms that the GTA evaluator is robust to different CoT parsing formats. It maintains high alignment with human judgment across diverse model outputs, as detailed in the consistency analysis in **Figure 3**.

---

### Official Review · Reviewer_LPGH · 2025-11-02

**Soundness:** 3
**Presentation:** 3
**Contribution:** 2
**Rating:** 4
**Confidence:** 3

**Summary:**

This paper introduces a diagnostic framework for diagnosing reasoning-execution gaps in VLM-powered mobile-use agents. The framework proposes the Ground-Truth Alignment (GTA) metric to assess whether the action implied by chain-of-thought (CoT) reasoning aligns with the ground-truth action. By combining GTA with the standard Exact Match (EM) metric, this diagnostic framework categorizes model outputs into four quadrants: (i) Ideal, where both reasoning and action are correct; (ii) Execution Gap (EG), where reasoning is correct but execution fails; (iii) Both Wrong, where both reasoning and action are incorrect; and (iv) Reasoning Gap (RG), where the action is correct but reasoning is inconsistent. In addition, the paper presents an automatic GTA evaluator validated through stratified sampled human annotation, and conducts experiments on three mobile-interaction benchmarks (AITZ, CAGUI, and AndroidControl) using multiple state-of-the-art models. Results reveal that reasoning-execution gaps are prevalent, with execution gaps occurring more frequently than reasoning gaps. Besides, causal CoT models typically exhibit RG > EG. Most models are capable of correct reasoning, at least in generating valid thought chains, but encounter difficulties in mapping these intermediate reasoning steps to the precise executable actions required by the task. Furthermore, scaling up model size reduces the overall gap but does not eliminate it.

**Strengths:**

This paper investigates an less-studied aspect of faithfulness in mobile-use agents: the inconsistency between CoT and its executed actions. In addition, the proposed GTA metric and four-quadrant framework provide a novel and principled method to disentangle reasoning accuracy from execution accuracy, surpassing the conventional EM metric in interpretability. The automatic GTA evaluator, built upon AgentCPM-GUI-8B and validated through human annotations, achieves accuracies ranging from 78% to 93% across diverse datasets and models, enabling scalable analysis without extensive manual effort. Experimental results across various benchmarks and model families (e.g., AgentCPM, UI-TARS, GUI-Owl) further demonstrate that models face a bottleneck in correctly executing the outcomes of CoT reasoning. Overall, this work demonstrates originality by offering a clear categorization within the diagnostic framework for mobile-use agents, while intuitively highlighting the primary and secondary issues in existing mobile-use agents, thereby informing future enhancements in agent reliability.

**Weaknesses:**

1. The work primarily emphasizes diagnosis without proposing methods to mitigate the identified gaps (e.g., through targeted fine-tuning), thereby leaving a gap between analytical insights and actionable improvements.

2. Certain visualizations, including charts and tables, exhibit presentation issues; for example, Figure 4 employs a line chart to illustrate differences among GTA, EM, and IDEAL, which may mislead readers due to implied trends—replacing it with a bar chart or another non-trend-oriented format would enhance clarity.

3. Compared to Execution Gap (EG), the Reasoning Gap (RG) addressed in the paper may not represent a highly significant research focus, potentially diminishing the work's contribution. Because this seems to be a relatively minor point in mobile-use agents.

**Questions:**

1. I would like to see some execution examples of the GTA Evaluator, including both successful and erroneous cases, to provide readers with a clearer understanding of how GTA operates.
2. Why does Figure 9 in the Appendix fall into the RG category? What errors are present in the model’s CoT results? Could you explain the rationale behind Figure 9 or provide additional examples of RG for further clarity?
3. The evaluation result appears to emphasize three intuitively plausible categories—EG, Ideal, and Both Wrong—as the most prevalent scenarios. In contrast, the RG scenario, presented in the diagnostic framework, seems less critical in practice. If the authors could provide additional examples and in-depth analysis of RG to better underscore its significance, I would consider raising my score.

---

> ### Author Response · Authors · 2025-12-03
>
> Dear Reviewer LPGH:
>
> **Thanks for your insightful review and constructive feedback.**
>
> > W1: The work primarily emphasizes diagnosis without proposing methods to mitigate the identified gaps (e.g., through targeted fine-tuning), thereby leaving a gap between analytical insights and actionable improvements.
>
> **Reply:** We added Section 4.4 to analyze scaling effects. Additionally, we tested a two-step CoT strategy with UI-TARS. As shown in the table, while CoT improved success rates, it notably failed to reduce EG/RG, highlighting that these gaps are persistent challenges requiring architectural innovation beyond simple prompting.
>
> | **Experimental Setting**         | **Method / Strategy**    | **EG (↓)** | **RG (↓)** | **Success Rate (↑)** |
> | -------------------------------- | ------------------------ | ---------- | ---------- | -------------------- |
> | **1. Scaling Effects** (Sec 4.4) | UI-TARS-2B               | 13.13%     | 3.42%      | 69.69%               |
> |                                  | UI-TARS-7B               | 12.81%     | 3.23%      | 72.18%               |
> |                                  | **UI-TARS-72B (Scaled)** | **12.17%** | **2.22%**  | **73.14%**           |
> | **2. Mitigation Strategy**       | UI-TARS-1.5-7B           | 5.53%      | 5.41%      | 75.07%               |
> |                                  | **+ Two-step Plan**      | 5.65%      | 5.45%      | **76.08%**           |
>
> > W2: Certain visualizations, including charts and tables, exhibit presentation issues; for example, Figure 4 employs a line chart to illustrate differences among GTA, EM, and IDEAL, which may mislead readers due to implied trends—replacing it with a bar chart or another non-trend-oriented format would enhance clarity.
>
> **Reply:** Thank you for the suggestion. We have replaced Figure 4 with a grouped bar chart. This format avoids implying non-existent trends and clearly highlights the distinct performance differences among GTA, EM, and IDEAL, ensuring better clarity.
>
> > Q2: Why does Figure 9 in the Appendix fall into the RG category? What errors are present in the model’s CoT results? Could you explain the rationale behind Figure 9 or provide additional examples of RG for further clarity?
>
> **Reply:** In Figure 9, the CoT correctly derives a visual plan ("click top-left"), but the model ignores it to execute a disconnected system instruction (`PRESS(back)`). This demonstrates that the reasoning process failed to causally guide execution, fitting our RG definition.
>
> > W3: Compared to Execution Gap (EG), the Reasoning Gap (RG) addressed in the paper may not represent a highly significant research focus, potentially diminishing the work's contribution. Because this seems to be a relatively minor point in mobile-use agents.
> >
> > Q3: The evaluation result appears to emphasize three intuitively plausible categories—EG, Ideal, and Both Wrong—as the most prevalent scenarios. In contrast, the RG scenario, presented in the diagnostic framework, seems less critical in practice. If the authors could provide additional examples and in-depth analysis of RG to better underscore its significance, I would consider raising my score.
>
> **Reply:** RG represents a critical safety bottleneck ("Silent Failure"). As shown in the new Figure 12, RG masks logic failures behind accidental success. This makes agents brittle and poses severe risks in safety-critical scenarios, rendering RG highly significant despite its lower statistical frequency.
>
> > Q1: I would like to see some execution examples of the GTA Evaluator, including both successful and erroneous cases, to provide readers with a clearer understanding of how GTA operates.
>
> **Reply:** We added Figures 13 and 14 to illustrate successful and erroneous cases. We observed that evaluator errors often stem from valid multi-solution paths in CoT that challenge strict matching. We are working on enhancing the evaluator to handle such open-ended scenarios.

---

### Meta-Review · Area_Chair_4Swx · 2025-12-04

**Summary:**

The reviewers' concerns primarily focused on the paper's limitation to diagnosis without providing mitigation strategies, the implementation details of the evaluation metric, and the robustness of the evaluator model.

Reviewer LPGH , Reviewer VPV5 , and Reviewer ScfG  all expressed concern that the work emphasized diagnosis while neglecting to propose novel methods to close the gaps. A significant technical concern raised by Reviewer 2Hkw and Reviewer ScfG  was the lack of clarity regarding the GTA Evaluator's implementation, specifically questioning if it required fine-tuning and how it handled biases. Furthermore, Reviewer LPGH questioned the significance of the Reasoning Gap compared to the Execution Gap, while Reviewer ScfG  strongly criticized the reliance on a single evaluator model, arguing it might reflect evaluator bias rather than ground truth.

**Reviewer Concerns:**

Regarding the lack of mitigation, the authors responded to Reviewer LPGH and VPV5 by adding an analysis of scaling effects and testing two-step CoT strategies, empirically showing that these gaps persist even with these interventions, thereby justifying the need for their diagnostic framework.

The confusion regarding the GTA Evaluator's implementation was resolved for Reviewer 2Hkw and ScfG by clarifying that it is a training-free, zero-shot module.

To address Reviewer ScfG's concern about single-model bias, the authors provided new cross-model validation results using different backbones, demonstrating consistent agreement with human ground truth.

While the core validity concerns were addressed, the absolute accuracy of the evaluator in some datasets, as noted by Reviewer ScfG , remains an inherent challenge of model-based evaluation, though the authors provided explanations for this.

**Reviewer Scores:**

Reviewer LPGH would likely raise their score, as the authors directly addressed their specific questions about the significance of the Reasoning Gap (RG) with new case studies and clarified the mitigation aspect with new experiments. Reviewer 2Hkw would also likely increase their score, as their primary technical critique stemmed from a misunderstanding about the evaluator requiring fine-tuning, which the authors explicitly corrected as being a zero-shot implementation. Reviewer VPV5's concerns regarding history bias and the need for case studies were addressed, likely leading to a maintained or slightly improved score. Reviewer ScfG, who gave the lowest score, would likely raise the score, given that the authors provided the critical cross-model validation data requested to prove the findings were not merely artifacts of evaluator bias.

 However, despite these potential score increases, the fundamental consensus regarding the paper's limited contribution, specifically its focus on diagnosis without methodological innovation, remains a bottleneck. Given that the starting scores were relatively low (mostly 4s and a 2), even with credit for the rebuttal, the final consensus might still hover marginally below the strict acceptance threshold due to the perceived lack of technical novelty.

---

### Decision · Program_Chairs · 2026-01-26

Reject